# Genetically Elevated Selenoprotein S Levels and Risk of Stroke: A Two-Sample Mendelian Randomization Analysis

**DOI:** 10.3390/ijms26041652

**Published:** 2025-02-14

**Authors:** Yan He, Yi Liu, Haoliang Meng, Jinsheng Sun, Yukun Rui, Xiaoyi Tian, Zhengbao Zhu, Yuzhen Gao

**Affiliations:** 1Jiangsu Key Laboratory of Preventive and Translational Medicine for Geriatric Diseases, Department of Epidemiology, School of Public Health, Suzhou Medical College, Soochow University, Suzhou 215006, China; yhe@suda.edu.cn (Y.H.); liuyi201016@163.com (Y.L.); 2Department of Forensic Medicine, Suzhou Medical College, Soochow University, Suzhou 215006, China; 20234221083@stu.suda.edu.cn (H.M.); sunjingsheng2025@163.com (J.S.); r_yukun18@163.com (Y.R.); 3School of Public Health, Dalian Medical University, Dalian 116041, China; tianxiaoyi1218@163.com

**Keywords:** selenium, stroke, Mendelian randomization, SELENOS, intracerebral hemorrhage

## Abstract

Selenoprotein S (SELENOS), one of the carrier proteins of dietary selenium (Se), is a key regulator of inflammation, oxidative stress, and endoplasmic reticulum (ER) stress, all of which are implicated in the pathogenesis of stroke. However, the causality between SELENOS and stroke risk remains poorly understood. This study aimed to explore the association between genetically determined plasma SELENOS levels and the risk of all-cause stroke, ischemic stroke, and intracerebral hemorrhage (ICH) using a two-sample Mendelian randomization (MR) approach. We analyzed data from three large-scale Genome-Wide Association Study (GWAS) meta-analyses of individuals of European descent. The fixed-effect inverse-variance weighted (IVW) model analysis revealed that genetically elevated SELENOS levels were associated with an increased risk of all-cause stroke, ischemic stroke, and ICH. Sensitivity analyses showed no evidence of pleiotropy or heterogeneity, and leave-one-out analyses confirmed the robustness of our results. Here, we show that elevated plasma SELENOS levels are causally linked to increased stroke risk. Although the effect sizes were modest, these findings suggest SELENOS may play a role in stroke pathogenesis, emphasizing the need for further mechanistic and functional studies. Finally, our findings shed light on the importance of tailored Se intake management in the context of stroke prevention.

## 1. Introduction

Stroke is the second-leading cause of death globally and the third-largest contributor to disability worldwide [1,2]. Although stroke rates have declined in high-income countries over the past three decades, they have risen in low- and middle-income countries due to changes in epidemiological, socioeconomic, and demographic factors, resulting in an overall increase in the global burden of stroke [3,4]. The development of stroke involves complex pathophysiological pathways, and traditional risk factors only explain a portion of its overall risk. As a result, it is crucial to identify novel biomarkers to better understand its underlying mechanisms and uncover potential therapeutic targets [5,6].

Selenium (Se), an essential trace element, plays a crucial role in various biological functions, primarily through its incorporation into selenoproteins [7]. Se’s impact on human health has been extensively studied, with evidence suggesting that adequate Se intake may reduce the risk of atherosclerosis, coronary artery disease, and stroke [8,9,10]. Low selenium levels have been linked to increased oxidative damage, inflammation, and endothelial dysfunction, which contribute to the development of cardiovascular diseases [11,12,13]. Conversely, some studies have shown that excessive Se levels, often from supplementation, may have adverse effects, including promoting oxidative stress and inflammation, leading to a potential increase in cardiovascular disease risk [14,15]. This dual nature of Se—beneficial at optimal levels but potentially harmful at high concentrations—underscores the importance of balanced selenium intake for cardiovascular health.

Selenoprotein S (SELENOS), one of the selenoprotein family members, plays a crucial role as a metabolite of dietary Se [16,17]. It enhances the catalytic effects of Se and is characterized by its unique molecular structure, which contains selenocysteine (Sec)—a selenium-based amino acid encoded by the stop codon UGA [18,19]. It is well documented that SELENOS plays a pivotal role in regulating inflammation, oxidative stress, and endoplasmic reticulum (ER) stress [20,21,22,23,24], all of which typically exhibit reciprocal causation, mutually reinforcing each other, and collectively contribute to the onset and progression of stroke. Previous studies based on cross-sectional/case–control design have indicated that the circulating Se level may inversely associated with risk of stroke, although further evidence is still needed to conclude their definite association [25,26]. According to a recent observational study, dietary selenium also showed a negative, non-linear correlation with stroke risk in adults, with variations observed among different population subgroups [27]. Additionally, numerous previous studies genetic variations within the SELENOS gene locus correlates with ischemic stroke and cardiovascular disease [28,29,30].

In this regard, we hypothesize that plasma SELENOS levels may be associated with risk of stroke. In order to investigate our hypothesis, we assessed the correlation between SELENOS levels, as genetically indicated, and the likelihood of stroke occurrence. We utilized Mendelian randomization (MR), a widely recognized method for discerning causal links between exposures and health outcomes, leveraging genetic variants—specifically single-nucleotide polymorphisms (SNPs)—as instrumental variables (IVs). This approach allows for the examination of potential causality while minimizing the influence of confounding factors [31]. As these genetic variations are determined before birth, MR estimates remain unaffected by reverse causation or confounding variables, thus circumventing the constraints of observational studies [32]. In the current study, a bidirectional MR analyses (switching the positions of exposures and outcomes) was implemented to perform the analysis using summary statistics from the largest publicly available GWAS for SELENOS and stroke. Subsequent rigorous tests for heterogeneity, pleiotropy, and sensitivity were performed to inspect robustness of the analysis.

## 2. Results

### 2.1. Attributes of the Genetic Instruments and Their Statistical Power Assessment

The study design for the current study is depicted in Figure 1. In this MR investigation, a collection of 15 distinct SNPs, mapped to chromosome 19, were designated as the genetic instruments for SELENOS (Table 1). These SNPs accounted for 36.53% of the variability in SELENOS plasma levels, indicating their substantial influence on the phenotype. The calculated average F statistic for this set of genetic instruments was 89.99, which is well above the threshold that indicates a low probability of weak instrument bias influencing this study’s findings. Furthermore, the power analysis conducted for our MR study indicated that we had an adequate-to-high capacity to detect the linkage between SELENOS levels and stroke outcomes (Appendix A).

### 2.2. Associations Between Stroke Risk and SELENOS Levels as Determined by Genetic Factors

Based on the Cochran Q test results, no significant heterogeneity was detected among the genetic instruments (*p* values exceeding 0.05). For the analysis of all-cause stroke, ischemic stroke, and ICH, fixed-effects IVW models were applied, as detailed in Appendix A. The findings, depicted in Figure 2, indicated a positive correlation between genetically predicted plasma SELENOS levels and the risk of all-cause stroke (OR per SD increase, 1.035 [95% CI, 1.012–1.059]; *p* = 0.003), ischemic stroke (OR per SD increase, 1.035 [95% CI, 1.010–1.061]; *p* = 0.006), and ICH (OR per SD increase, 1.129 [95% CI, 1.056–1.207]; *p* = 0.0004). Further exploratory subgroup analyses revealed that elevated genetically predicted plasma SELENOS concentrations were linked to a higher risk of large-artery stroke (OR per SD increase, 1.077 [95% CI, 1.004–1.156]; *p* = 0.038, Appendix A). No significant connections were identified between SELENOS levels and the risks of cardioembolic stroke (OR per SD increase, 0.969 [95% CI, 0.916–1.025]; *p* = 0.272) or small-vessel stroke (OR per SD increase, 1.026 [95% CI, 0.969–1.086]; *p* = 0.378). Figure 3 provides a graphical representation of the associations between each genetic variant and plasma SELENOS levels, along with the related risks for all-cause stroke, ischemic stroke, and intracerebral hemorrhage (ICH).

### 2.3. Sensitivity Analyses

A range of MR sensitivity analyses models was employed to examine the robustness of our study’s results, as outlined in Table 2. Sensitivity analyses with the weighted median, MR-RAPS, MR-PRESSO, and MR-Egger methods produced results consistent with the main analyses, except for the weighted median MR and MR-Egger analyses, where the association between SELENOS levels and ischemic stroke, as well as ICH, was weakened. As shown in Table 3, the MR-PRESSO global test did not detect any signs of horizontal pleiotropy, and the intercept from the MR-Egger regression confirmed no evidence of directional pleiotropy, supporting the validity of the instrumental variable assumptions. Additionally, the leave-one-out analysis demonstrated that none of the SNPs had a dominant impact on the observed associations, as illustrated in Appendix A. The directional Steiger test further verified the correctness of these associations (all *p* < 1.67 × 10^−2^, Appendix A). In contrast, the bidirectional IVW MR analysis did not uncover significant links between genetically predicted stroke outcomes and SELENOS levels (Appendix A).

Collectively, this two-sample MR study provides evidence that genetically elevated plasma levels of SELENOS are associated with an increased risk of all-cause stroke, ischemic stroke, and ICH (Figure 2). The findings were robust across various sensitivity analyses, including MR-PRESSO and leave-one-out analyses, which confirmed the absence of significant pleiotropy or heterogeneity and indicated that no single instrumental variable disproportionately influenced the results (Table 2 and Table 3). Subgroup analyses further revealed a significant association between higher SELENOS levels and an increased risk of large-artery stroke, while no significant links were found for cardioembolic or small-vessel stroke (Appendix A).

## 3. Discussion

In this two-sample MR study utilizing data from large-scale GWASs, our results revealed that genetically elevated plasma SELENOS levels were linked to a higher risk of all-cause stroke, ischemic stroke, and intracerebral hemorrhage (ICH). Sensitivity analyses, such as MR-PRESSO, showed no signs of horizontal pleiotropy, and the leave-one-out analysis confirmed the reliability of these findings, indicating that no single instrumental variable was responsible for the observed associations.

SELENOS, a key player in cellular homeostasis, plays dual roles in stroke pathology, influenced by its impact on oxidative stress, inflammation, and ER stress. SELENOS is recognized for its anti-inflammatory and antioxidant properties, which are crucial in maintaining vascular homeostasis [33,34]. The inflammatory process is a key component in the pathogenesis of atherosclerosis, a major risk factor for ischemic stroke [35]. SELENOS has been found to decrease the expression of pro-inflammatory cytokines like tumor necrosis factor-alpha and interleukin-6, which may help reduce the risk of atherosclerotic events [36]. Similarly, its antioxidant function could mitigate oxidative stress, a pathophysiological mechanism implicated in cerebral ischemia and hemorrhage [37]. However, our findings support a dual role for SELENOS, where at certain levels, it may contribute to stroke risk rather than being solely protective. In line of this, this U-shaped relationship between SELENOS levels and stroke risk could be analogous to the non-linear association observed between selenium intake and the risk of stroke and all-cause mortality in observational studies [38,39,40,41]. It is plausible that an optimal range of SELENOS levels is necessary for its protective effects, with deviations from this range potentially leading to adverse outcomes. It can also be assumed that a subtle balance of Se metabolism in the body is critical for overall health and stroke prevention.

The primary strength of this study lies in its use of a two-sample MR design, which helps to mitigate confounding and reverse causation [42,43], thereby providing stronger evidence for a causal relationship between SELENOS levels and stroke risk. Additionally, this study’s robust statistical analyses, including sensitivity tests like MR-PRESSO and leave-one-out analysis, enhance the validity of the findings. However, there are limitations to consider. The genetic instruments used to proxy SELENOS levels, while robust, may not capture the full spectrum of SELENOS activity in the body [44,45,46]. Furthermore, the generalizability of these findings may be limited to populations of European ancestry, as the genetic variants used in the analysis were predominantly derived from this group. Considering the ethnic diversity in SELENOS genetics and the complexity of its role in stroke risk, additional replicative follow-up studies in other populations are needed to validate the current findings. Finally, while MR analysis may reduce the risk of confounding, it cannot entirely eliminate it, especially in the presence of unmeasured pleiotropy [47,48]. Although our MR analysis suggests a statistically significant association between genetically elevated SELENOS levels and increased risk of stroke, the small effect sizes (ORs close to 1) indicate that this relationship is likely of limited clinical or epidemiological relevance. Further research is needed to determine whether SELENOS plays a meaningful role in stroke pathogenesis or if it is merely a minor contributing factor.

The findings from this study suggest that SELENOS could serve as a potential biomarker for stroke risk. For example, measuring SELENOS protein levels in the blood could serve as an early indicator of stroke risk. SELENOS-based therapies could target its regulation to optimize protective effects while minimizing potential harm. This could have significant implications for the development of targeted prevention strategies. For instance, individuals with genetically elevated SELENOS levels might benefit from more aggressive management of traditional stroke risk factors, such as hypertension and hyperlipidemia [49,50]. Moreover, these results highlight the importance of balanced selenium intake, as SELENOS is a key metabolite of dietary selenium. The study cautions against excessive selenium supplementation, which could potentially elevate SELENOS levels and thereby increase stroke risk, emphasizing the need for precise nutritional recommendations tailored to individual needs [51]. Further research is warranted to explore the therapeutic potential of modulating SELENOS levels in stroke prevention and to determine whether these findings can be generalized to other populations.

Finally, this research opens several potential avenues for future studies. Replicating these findings in diverse populations, particularly those of non-European descent, is essential for understanding the generalizability of the results. Furthermore, research is needed to elucidate the precise molecular mechanisms through which SELENOS influences stroke risk, including its interactions with other metabolic pathways and environmental factors. Future investigations should also explore the therapeutic potential of modulating SELENOS levels, either through dietary adjustments or targeted interventions, to assess its practical applicability in stroke prevention. This study serves as a foundation for further exploration of SELENOS’s role in stroke and other vascular diseases, providing an impetus for future research that could ultimately inform clinical strategies and improve health outcomes on a broader scale.

## 4. Materials and Methods

### 4.1. Study Design

The data that support the findings of this study are available from the corresponding author upon reasonable request. This research adheres to the Strengthening the Reporting of Observational Studies in Epidemiology Using Mendelian Randomization (STROBE-MR) guidelines [52]. We conducted a two-sample MR analysis to explore the potential role of plasma SELENOS in stroke development, including different subtypes. The MR approach is based on three key assumptions: (1) the genetic variants should significantly influence the exposure of interest, (2) the genetic variants should be independent of potential confounding factors, and (3) the genetic variants should affect the outcome solely through their impact on the exposure. We identified genetic instruments that are indicative of SELENOS levels and used these as proxies to assess their impact on stroke risk. The data for SELENOS levels were sourced from a previous GWAS [53], while the stroke outcome data including all-cause stroke and ischemic stroke (IS) were obtained from the MEGASTROKE consortium and additional GWAS meta-analyses, focusing on individuals of European descent [54]. Additional data regarding intracerebral hemorrhage (ICH) were sourced from a meta-analysis of GWAS conducted in collaboration with the UK Biobank and FinnGen [55]. For the purposes of ensuring scientific integrity and facilitating collaborative research, the underlying data supporting the findings of this investigation are readily obtainable from the primary investigator upon a legitimate request. It is worthy of note that the current MR analysis exclusively utilized publicly accessible aggregate data from prior GWASs, thereby rendering the requirement for ethical approval unnecessary. This is because the foundational studies from which the data were derived have previously secured the necessary approvals from ethical review boards and have obtained the informed consent of all participants involved.

### 4.2. Genetic Instruments Selection for Plasma SELENOS Levels

For the identification of genetically relevant instrumental SNPs, we initiated our selection process by filtering for SNPs that achieved a genome-wide significance threshold with a *p*-value of less than 5 × 10^−8^. Subsequent to this filtering, we applied a clumping process to ensure that the selected SNPs were independent, using a linkage disequilibrium (LD) threshold of r^2^ less than 0.1 within a 10,000 kilobase pair window. We then eliminated SNPs that were not represented in the GWAS summary data for the outcomes of interest and also excluded palindromic SNPs with a minor allele frequency exceeding 0.4 to harmonize the data for exposure and outcome. In addition, if the SELENOS-associated SNP was not available in the MEGASTROKE dataset, a proxy SNP (r^2^ > 0.8) was selected by default based on a 1000 Genomes European reference panel. *F*-statistic was used to estimate the strength of the genetic instruments for SELENOS and stroke. The calculation formula was as follows: *F* = (*N* − *K* − 1) × *R*^2^/*K* × (1 − *R*^2^). *R*^2^ is the phenotypic variance explained by the genetic variants, N is the sample size of the exposure GWAS, and *K* is the number of the genetic variants. An *F*-statistic greater than 10 indicates a strong instrument [56]. As a result of this stringent selection process, we identified 15 SNPs that were significantly associated with plasma SELENOS levels, with all of these SNPs mapping to chromosome 19 (Table 1).

### 4.3. Data Sources for Outcomes

Aggregate data for multiple stroke categories, encompassing all-cause stroke incidence and ischemic stroke (IS), were extracted from the MEGASTROKE initiative. This initiative encompassed a comprehensive meta-analysis that integrated findings from 29 GWAS studies focused on individuals of European heritage [54]. The dataset included a total of 40,585 individuals who had experienced a stroke and a control group of 406,111 individuals. Within the MEGASTROKE dataset, there were 34,217 instances of ischemic stroke with diagnoses confirmed through clinical assessments and imaging profiles. Furthermore, statistical data pertaining to intracerebral hemorrhage (ICH) were sourced from a meta-analysis conducted in collaboration with the UK Biobank and FinnGen [55]. This analysis included 1935 subjects with ICH and a control group of 471,578 subjects, all of whom were of European descent. The ICH cases were identified through brain imaging that confirmed the presence of bleeding within the brain tissue.

### 4.4. Statistical Analysis

In the context of this research, the primary analytical approach was the fixed-effect inverse-variance weighted (IVW) model. This model is used to combine estimates across SNPs and is commonly recognized for its effectiveness in causal inference scenarios, especially in the absence of directional pleiotropy [57]. Within the scope of our MR analysis, the results were quantified in terms of odds ratios (ORs) and their associated 95% confidence intervals (CIs). These were determined for the risk associated with all-cause stroke, IS, and ICH, corresponding to a standard deviation increment in the genetically associated log-transformed plasma levels of SELENOS. To evaluate the heterogeneity across the genetic instruments, the Cochran Q statistic was applied [58].

To reinforce the validity of our findings and account for potential deviations from MR assumptions, we performed an array of sensitivity analyses. These included the weighted median estimation, the MR robust adjusted profile score (MR-RAPS), the MR pleiotropy residual sum and outlier (MR-PRESSO) test, MR-Egger regression, and a leave-one-out approach. Weighted median estimation can provide consistent estimates when some of the genetic variants in the analysis are not valid IVs [59]. The MR-RAPS framework, which incorporates a random-effects model, was used to detect and adjust for pleiotropic effects of the genetic variants, thereby mitigating the impact of horizontal pleiotropy and weak instrument bias. MR-PRESSO employs a “leave-one-out” methodology to assess whether a specific SNP instrument is responsible for the observed differences in computed residual sum of squares compared to simulated expectations [60]. This model was applied to identify and correct for any outliers among the genetic instruments, enhancing the reliability of our estimates. Additionally, we utilized MR-Egger regression to evaluate the overall pleiotropic effects by examining the intercept, providing insight into the directional pleiotropy present across all SNPs [61]. To ensure the stability of our MR findings, we conducted a leave-one-out analysis, sequentially excluding each SNP to assess its individual influence on the results [62]. For statistical significance, we applied a stringent threshold of *p* < 1.67 × 10^−2^, adjusted for multiple testing using the Bonferroni method, which involved dividing the standard alpha level of 0.05 by the number of comparisons (three in this case, for one exposure and three outcomes). In the subsequent exploratory subgroup analyses, statistical significance was defined as a *p*-value of less than 0.05.

The determination of the statistical power for this MR study was facilitated through the utilization of the mRnd online platform. For conducting the analytical procedures, we leveraged the ’TwoSampleMR’ and ’MR-PRESSO’ packages within the R programming environment, specifically version 4.2.1, as provided by the R Development Core Team.

## 5. Conclusions

In summary, this study provides initial genetic evidence linking genetically elevated plasma SELENOS levels to an increased risk of all-cause stroke, ischemic stroke, and ICH, suggesting a potential role for SELENOS in stroke development. By utilizing a two-sample Mendelian randomization approach, we minimize confounding and reverse causation, offering robust insights into the role of SELENOS in stroke pathogenesis. The findings underscore the importance of maintaining optimal levels of SELENOS for vascular health and shed light on the importance of balanced selenium intake in the context of stroke risk. Although the associations are statistically significant, the modest effect sizes indicate that SELENOS has limited predictive value for stroke risk. This underscores the need for further research to clarify whether SELENOS plays a meaningful role in stroke pathogenesis or is merely a minor contributing factor. Furthermore, the generalizability of these findings to non-European populations and the need for further mechanistic studies remain key areas for future research. As such, this study opens new avenues for targeted prevention strategies and further investigation into SELENOS’s therapeutic potential, solidifying its place as a critical focus for future stroke-related research.

## Figures and Tables

**Figure 1 ijms-26-01652-f001:**
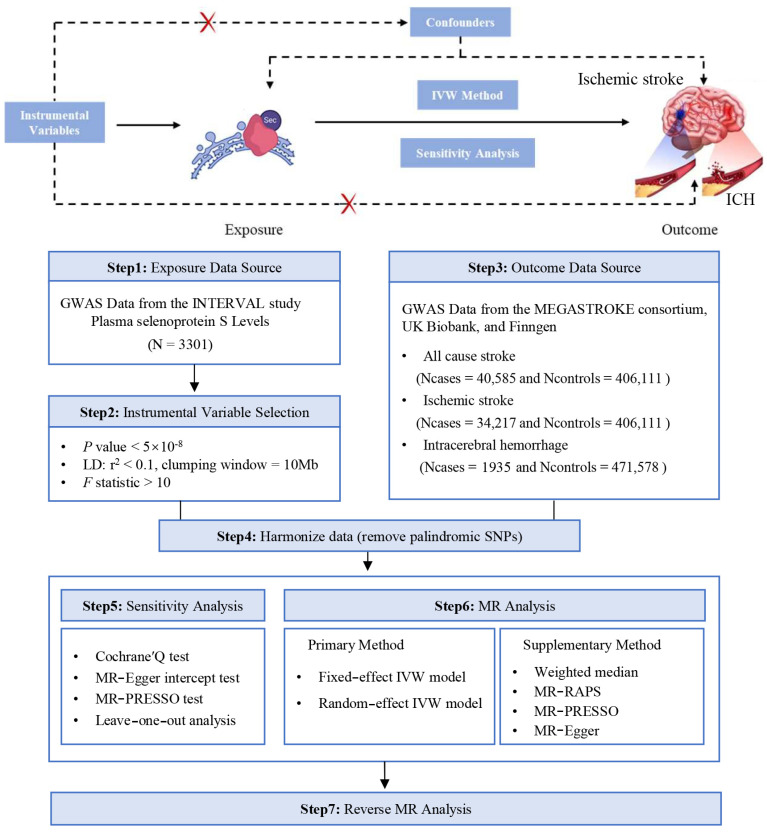
Study design for the Mendelian randomization analysis of plasma SELENOS levels and the risks of stroke.

**Figure 2 ijms-26-01652-f002:**
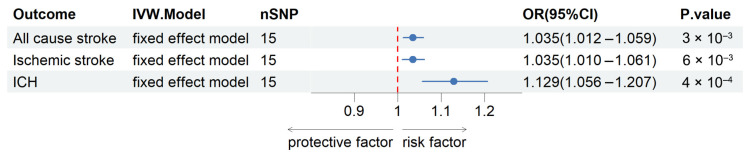
Forest plots for the associations of plasma SELENOS levels with risks of all-cause stroke, ischemic stroke, and ICH in the main inverse-variance weighted Mendelian randomization analysis. OR, odds ratio; 95% CI, 95% confidence interval; SNP, single-nucleotide polymorphism; ICH, intracerebral hemorrhage.

**Figure 3 ijms-26-01652-f003:**
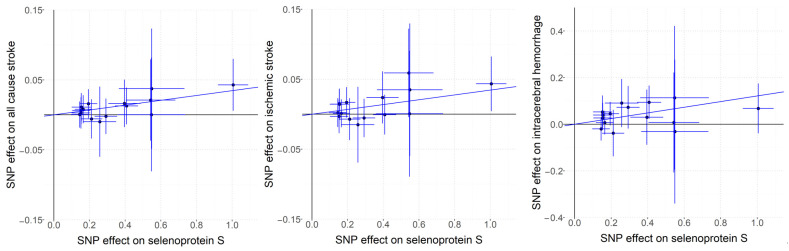
Scatterplots for associations of SELENOS variants with risks of all-cause stroke, ischemic stroke, and intracerebral hemorrhage (ICH).

**Table 1 ijms-26-01652-t001:** Characteristics of 15 genetic variants used as instrumental variables for plasma SELENOS levels.

SNP	Chr	Position (Build 37)	Nearest Gene	EA	OA	β	SE	*p* Value	*F* Statistic
rs60049679	19	45429708	APOC1	C	G	0.26	0.05	3.31 × 10^−8^	32.35
rs117261169	19	45491032	CLPTM1	T	C	−0.55	0.10	1.20 × 10^−8^	37.43
rs2965169	19	45251156	BCL3	C	A	−0.15	0.03	2.04 × 10^−9^	37.06
rs151330717	19	45196964	CEACAM16	A	G	−0.55	0.09	5.13 × 10^−9^	39.63
rs405509	19	45408836	APOE	G	T	−0.17	0.02	1.45 × 10^−11^	45.36
rs62117161	19	45233385	BCL3	G	A	−0.40	0.05	5.62 × 10^−18^	77.13
rs28399657	19	45318351	BCAM	G	A	−0.54	0.07	1.78 × 10^−14^	60.40
rs28399637	19	45324138	BCAM	A	G	0.15	0.03	5.62 × 10^−9^	33.87
rs7343130	19	45331103	BCAM	G	A	0.19	0.03	5.25 × 10^−15^	62.52
rs11668327	19	45398633	TOMM40	C	G	−0.29	0.03	8.32 × 10^−20^	84.70
rs4263041	19	45438643	APOC4	G	A	−0.21	0.03	6.92 × 10^−12^	64.10
rs429358	19	45411941	APOE	C	T	0.41	0.03	5.50 × 10^−35^	148.84
rs6859	19	45382034	NECTIN2	G	A	−0.15	0.03	5.01 × 10^−9^	34.10
rs7412	19	45412079	APOE	T	C	−1.01	0.04	1.78 × 10^−120^	559.13
rs4803759	19	45327459	BCAM	C	T	0.15	0.03	1.35 × 10^−8^	33.26

SNP, single-nucleotide polymorphism; EA, effect allele; OA, other allele; SE, standard error.

**Table 2 ijms-26-01652-t002:** Sensitivity analyses for associations of plasma SELENOS levels with all-cause stroke, ischemic stroke, and ICH.

Outcome	NO. SNPs	Weighted Median		MR-RAPS		MR-PRESSO		MR-Egger	
		OR (95% CI)	*p* Value	OR (95% CI)	*p* Value	OR (95% CI)	*p* Value	OR (95% CI)	*p* Value
All-cause stroke	15	1.042 (1.011–1.075)	0.007	1.035 (1.011–1.060)	0.004	1.035 (1.017–1.053)	0.0001	1.040 (1.000–1.081)	0.052
Ischemic stroke	15	1.036 (1.001–1.072)	0.046	1.035 (1.010–1.062)	0.006	1.035 (1.011–1.060)	0.004	1.041 (0.998–1.085)	0.061
ICH	15	1.076 (0.980–1.181)	0.125	1.130 (1.056–1.210)	0.0004	1.129 (1.047–1.218)	0.002	1.095 (0.977–1.228)	0.119

*p* < 1.67 × 10^−2^ after Bonferroni correction (0.05/3 [1 exposure and 3 outcomes]) was considered to be statistically significant. ICH, intracerebral hemorrhage; MR-PRESSO, Mendelian randomization pleiotropy residual sum and outlier; MR-RAPS, Mendelian randomization robust adjusted profile score; OR, odds ratio; 95% CI, 95% confidence interval; SNP, single-nucleotide polymorphism.

**Table 3 ijms-26-01652-t003:** Pleiotropy assessment for associations of SELENOS levels with all-cause stroke, ischemic stroke, and ICH.

Outcome	MR-PRESSO Global Test		MR-Egger Intercept	
	Observed RSS	*p* Value	β (se)	*p* Value
All-cause stroke	4.89	0.99	−0.002 (0.006)	0.79
Ischemic stroke	10.51	0.84	−0.002 (0.006)	0.76
ICH	14.62	0.61	0.010 (0.016)	0.52

An observed 2-sided *p* value < 0.05 was considered nominal significance; ICH, intracerebral hemorrhage; MR-PRESSO, Mendelian randomization pleiotropy residual sum and outlier; RSS, residual sum of squares.

## Data Availability

All data utilized in this study are publicly accessible. Data are also available from the corresponding author upon reasonable request.

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
