# Peer review of "Genetically Elevated Selenoprotein S Levels and Risk of Stroke: A Two-Sample Mendelian Randomization Analysis"

_ijms, 2025, doi:10.3390/ijms26041652_

Round 1
Reviewer 1 Report
Comments and Suggestions for Authors
In this study, the authors attempted to explore the possible causal relationship between (arbitrary and apparently genetically influenced) plasma levels of selenoprotein S (SELENOS; anti-inflammatory, antioxidant) and the risk of all-cause stroke, ischemic stroke, and intracerebral hemorrhage (ICH) using three different genome-wide association study (GWAS) meta-analyses of European ancestry (identifying 15 potential SNPs (chromosome 19)) and analyzed by a rather complex statistical method (two-sample Mendelian randomization (MR); methods: MR-RAPS, MR-Presso, MR-Egger).
The experimental design and statistical analysis, although adequate for this type of exploratory analysis, generates highly questionable results and interpretations. For example: i) Regardless of the MR method used, the reported odds ratios vary between 1.036-1.095, indicating a negligible relationship (from an epidemiological point of view), which are supported by strangely very significant p values. ii) Between methods, the R value is equal or very close and in one method it appears significant and in another it does not. iii) Although the p-value allows assertions on the influence between studied factors, from a strictly epidemiological point of view it should be indicated that this relationship is practically null (low OR values) with the causes of mortality analyzed; authors are suggested to consider the above to modify their analysis and/or declare in their manuscript the low predictive value of SELENOS.
Other useful changes to consider are the following:
A) Improve grammar and syntax in English.
B) Describe the source of information more fully (although succinctly), describing image 1 step by step to measure the access process and the amount of data analyzed.
C) Extend the discussion on the limitations of the study at the end of the discussion section.
D) Include ethical approval
Comments on the Quality of English LanguageMajor editing is needed
Author Response
Please see the uploaded PDF file.

Reviewer 2 Report
Comments and Suggestions for Authors
From a biostats and clinical epidemiology point of view, the manuscript has been well planned and reported, only a couple of curiosities.
- you correctly performed an influential MA (i.e. leave-one-out), what about to perform a cumulative MA (i.e. adding any single statistical unit by their ascending date) too!? This could be useful to check if any time trend affect your conclusions
- what about further sensitivity analyses, in particular adding more subgroup ones!? Do you believe it would be advisable!?
Author Response
Please see the uploaded PDF file.

Round 2
Reviewer 1 Report
Comments and Suggestions for Authors
The reply letter and all the changes made to the manuscript are sufficient to continue with the editorial process. The manuscript has improved substantially.
Author Response
We are extremely grateful for your insightful and constructive comments on our manuscript. Your feedback is invaluable to us and undoubtedly will enhance the quality of our research.